# Emergence of Corona Is Independent of the Four Whorls of Floral Organs in *Narcissus tazetta*

**DOI:** 10.3390/plants12071458

**Published:** 2023-03-27

**Authors:** Yanjun Ma, Xiaomeng Hu, Keke Fan, Na Zhang, Lili Shang, Yayun Deng, Tao Hu, Wenbo Zhang, Yan Wang, Zehui Jiang

**Affiliations:** 1International Center for Bamboo and Rattan, Beijing 100102, China; mayanjun@icbr.ac.cn (Y.M.); xm766408356@126.com (X.H.); fankk@icbr.ac.cn (K.F.); zhangna402265@163.com (N.Z.); shangll@icbr.ac.cn (L.S.); yayundeng@icbr.ac.cn (Y.D.); hutao@icbr.ac.cn (T.H.); wenbozhang@icbr.ac.cn (W.Z.); 2Key Laboratory of National Forestry and Grassland Administration/Beijing for Bamboo & Rattan Science and Technology, Beijing 100102, China; 3Chinese Academy of Forestry Research Institute of Forestry, Beijing 100091, China

**Keywords:** Chinese narcissus, Jinzhanyintai, Yulinglong, stamen petalody, corona, morphogenesis, anatomy

## Abstract

Plants of the genus *Narcissus* are well-known for their characteristic corona morphology, which structural origins have been a bone of contention among scholars. With “Jinzhanyintai” (JZ) and “Yulinglong” (YLL)—two major close-originated cultivars of Chinese narcissus (*Narcissus tazetta* L. var. *chinensis* Roem)—as materials, anatomic observation was made on floral organs during corona morphogenesis by dissection with hands under a stereomicroscope, paraffin section, scanning electron microscopy, and high-resolution X-ray tomography. It was uncovered that corona primordia of both cultivars appeared following the end of the differentiation of other floral organs, with differentiation sites located at the inner wall of the juncture of the base of tepals and the upper margin of the hypanthium. Affected by staminal filaments, the corona primordia of JZ experienced a three-stage differentiation process, namely blockage from the second whorl of stamens, blockage from the first whorl of stamens, and healing of corona primordia. However, the expanded spatial structure of the first whorl of petal-like stamens blocked the path of differentiation of YLL corona primordia, giving rise to slow differentiation of the corona primordia at the base of the first whorl of petal-like stamens and malformed differentiation of the corona primordia in the interval between the two whorls of petal-like stamens. Thus, a fragmented structure consisting of typical and fragmented coronas was formed. Furthermore, petal-like stamens of YLL in the lower part had a corona-like morphology. The spatio-temporal specificity of corona differentiation convincingly demonstrates that the corona is a structure independent of and different from the typical four whorls of floral organs, but also highly correlated with stamen.

## 1. Introduction

Plants of the genus *Narcissus* are well-known for their characteristic corona morphology. The size and morphology of the *Narcissus* corona serve as one of the major bases for the development of international classification criteria for *Narcissus* cultivars by the Royal Horticultural Society [1]. For this reason, research on corona morphology and development is of particular importance.

Model plants such as *Arabidopsis thaliana* and *Petunia hybrid* have typical four concentric whorls of floral organs, with sepals, petals, stamens, and gynoecium in the order from outside to inside. As to *Narcissus*, the corona is obviously petal-like in terms of morphology, but its structural origins have been a bone of contention among scholars. Some scholars think that the corona is ligules of leaves [2], confluent petal stipules [3], modification of stamens [4,5] developed from the extension of tepals, or an intermediate structure between the petal and the anther [6]. However, another report demonstrated that the corona has no relation to stamens and develops from the tepal tube [7]. Waters et al. conducted expression analysis of floral organ characteristics genes and thus argued that the corona is more homologous to stamens than tepals despite its petal-like phenotype [8].

To probe into the origin of *Narcissus* corona, two cultivars of Chinese narcissus (*Narcissus tazetta* L. var. *chinensis* Roem), haplopetalous “Jinzhanyintai” (JZ) and multi-petalled “Yulinglong” (YLL) (Figure 1), were enrolled as subjects in this study. Both JZ and YLL have six milk-white petals, collectively known as tepals, without a calyx [9]. As to JZ, its corona is yellow, wineglass-shaped and inserted on the hypanthium. In addition, JZ has three stamens above and below, one gynoecium, three stigmas, and one inferior ovary. In terms of YLL, its corona is also yellow, but it is not ring-shaped but fragmented. Apart from that, YLL has six petal-like stamens in clusters instead of typical stamens, with a yellow base and a white upper part, and its gynoecium is vestigial [10]. Anatomical evidence of floral organs suggests that YLL is a modification from petalody of stamens of JZ [11].

Research on the development stage of the JZ corona has been briefly mentioned in several reports. For instance, Zhang et al. [12] assigned flower bud differentiation into seven stages based on paraffin sections of flower buds of JZ and believed that corona primordia are already formed when stamen primordia appear, i.e., the formation of corona primordia is almost simultaneous with the appearance of stamen primordia, which is similar to the findings of the study of *N. jonguilla* conducted by Shimada et al. [13] and the study of *N. tazetta* var. *chinensis* conducted by Kamae et al. [14]. In contrast, Chen [15] divided the floral organ differentiation of JZ into six stages and suggested that the corona does not appear until late September when gynoecium primordia have differentiated, consistent with the findings of Preece et al. [16] and Zhong [17]. Experimental methods are needed to further confirm whether the dispute over the conclusion is attributed to interspecific differences or sampling method errors.

The site and stage of JZ corona differentiation remain controversial, and fragmented abnormal mutations of the corona are observed during stamen petalody of YLL, with unclear mutation mechanisms at present. In this study, therefore, the stage and site of corona differentiation of JZ and YLL were observed by means of stereomicroscopy, paraffin section, scanning electron microscopy, and high-resolution X-ray tomography, offering important references for resolving the mechanism of corona morphogenesis.

## 2. Results

### 2.1. Anatomical Observation of the Corona at Each Differentiation Stage

The morphogenesis of the corona of JZ and YLL was observed and recorded under the stereomicroscope every six days from 31 July 2021.

As shown in Figure 2, no corona primordia were observed in period 6 (30 August). In this period, the hypanthium superior to the ovary of JZ and YLL began to elongate and grow, the YLL-6 stamen primordia completed the petalody successfully, and the JZ-6 stamens completed the morphogenesis basically, with three longitudinal grooves visible on the inner side of each of the six anthers and filaments beginning to elongate and develop, allowing easy stripping of stamens with dissecting forceps. Period 7 (5 September) was the early stage of corona differentiation, with small bulges appearing on the inner side of the hypanthium and the outer side of stamens or petal-like stamens of both JZ and YLL. These small bulges were suspected to be corona primordia in the process of differentiation. In period 8 (11 September), corona primordia of JZ-8 continued to differentiate and were continuously distributed in some parts, while the YLL-8 corona had continuous development and fragmented distribution, with more expanded corona primordia at the base of the second whorl of petal-like stamens. Periods 9–10 (17 September and 23 September) were the healing stage of the ring-like coronas in JZ, with a ring-like distribution of coronas found in 1 of the 30 cases in JZ-9 and observed in all cases in JZ-10, while the fragmented corona of YLL-10 rose in size. During periods 11–12 (29 September and 8 October), the JZ and YLL coronas continued to develop slowly, and 1 of the 30 cases of JZ-12 anthers turned yellow. During period 13 (13 October), all JZ-13 anthers turned yellow, implying the completion of the basic morphogenesis of all floral organs. During period 14 (25 October), approximately two weeks later, the corona stopped developing and showed a comparable phenotype to that in period 13. At this time, a ring of coronas with a gamogenic base, but a fragmented upper part, was conspicuously inserted into the inner side of the hypanthium following thorough stripping of YLL-14 tepals and petal-like stamens.

### 2.2. Anatomical Observations of the Morphogenetic Site of Coronas

As shown in Figure 3 and Figure 4, the paraffin sections of JZ and YLL coronas in differentiation and development in periods 6–12 further validated the stereomicroscopic observations on coronas at the differentiation stage, and differentiation sites of coronas were observed with more details.

According to Figure 3, the anthophore in JZ-6 began to elongate and grow, the ovary and stigma of gynoecium (gy) began to differentiate and form, the basic form of anthers appeared, and the hypanthium (hy) had completed differentiation. In JZ-8, the ovary was clearly dilated, the ovule was basically formed, hy elongated and grew significantly, and the protruded corona (co) primordia appeared significantly, indicating that co differentiation is later than tepal (te), stamen (st), and gy differentiation and almost synchronizes with hy elongation and growth. In JZ-9, JZ-10, and JZ-12, co primordia gradually elongated and developed, accompanied by the elongation of hy. In JZ-9, pollen cell clusters were found to begin to appear in anthers, and in JZ-12, they were formed basically, suggesting gradual maturation of stamens.

In YLL-6, the ovary and stigma were conspicuously deformed, and the stigma was petal-like, but there were still a few ovules in the ovary (Figure 4). In addition, small bulges suspected to be corona (Co) primordia were found between the base of tepals (te) and petal-like stamens (ST). In YLL-8 (the longitudinal section of the second whorl of petal-like stamens of YLL), the base of the second whorl of petal-like stamens was located at the lower part of the hypanthium, and the morphogenetic site of Co primordia was not disturbed by petal-like stamens and was longer than that of JZ-8 in the same period. The base of the first whorl of petal-like stamens of YLL was inserted at the top of the hypanthium, with the Co primordia just present, indicating that its differentiation and development processes are apparently disturbed by the first whorl of petal-like stamens.

Moreover, scanning electron microscopy was carried out to further observe the morphology of floral organs of JZ and YLL in periods 8, 10, and 12 of corona morphogenesis.

Figure 5 shows the morphology of the floret after stripping tepals (te) and stamens (st) in JZ-8. At this time, the corona (co) primordia began to differentiate on the hypanthium (hy) between te and st, with more visible co primordia on the inner side of the gap between adjacent te, and the co primordia at the juncture of te and st were not yet separated from them. In JZ-10, a continuous ring of co primordia was clearly visible. In the longitudinal section of JZ-10z, co and te were inserted in the inner and outer sides of the top of the elongated hy. In JZ-12, co completed morphogenesis basically, but in JZ-12b, a small notch was found on the upper margin of co at the filament fi of the first whorl of st, while no notch was observed at the fi of the second whorl of st (JZ-12a). This is probably because the co differentiation path was blocked by the blockage from the first whorl of st, but such a blockage was not permanent. It was found in JZ-12z that cell inclusions at the point where the hy tip bound to the co were enormously rich, suggesting that this point is in an active phase of cell differentiation and proliferation.
Figure 5Scanning electron microscopic images of the JZ corona during morphogenesis. hy—hypanthium; gy—gynoecium; te—tepal; st—stamen; co—corona; fi—filament. JZ-8 represents all tepals and stamens stripped for JZ in period 8; JZ-10 represents all tepals stripped for JZ in period 10; JZ-10z and JZ-12z represent longitudinal sections for JZ in periods 10 and 12, respectively; JZ-12a and JZ-12b represent the second and the first whorl of stamens for JZ in period 12, respectively.Similar to that in JZ-8, irregular bulges suspected to be corona (Co) primordia were found on the inner side of the gap between tepals in YLL-8 (Figure 6). In YLL-10, the Co primordia at the juncture of the base of the second whorl of petal-like stamens (ST2) and the tepal (te) developed into a corona similar to that of JZ in morphology, with half surrounded at the base of ST2, but the development of the Co at the base of the first whorl of petal-like stamens (ST1) was clearly later than that of Co of ST2, consistent with the findings observed in paraffin sections. Hence, there were two different morphologies of Co primordia in YLL-10z: typical Co1 in ST2 and fragmented Co2 in ST1. In YLL-12z, Co further developed, and its longitudinal morphology was almost the same as that of the JZ co.
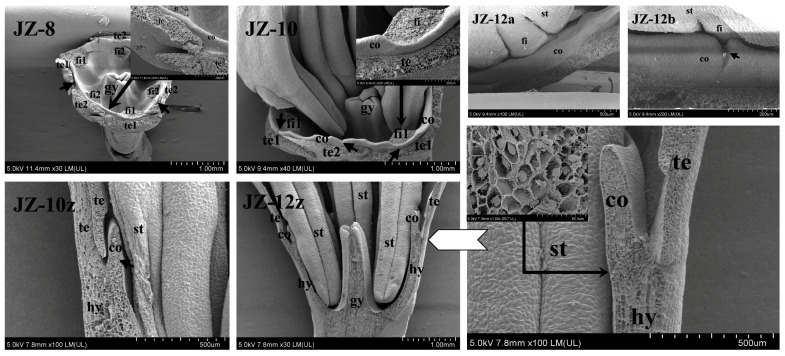


Furthermore, 360°-view high-resolution X-ray tomography was conducted for the florets at the early stage of corona morphogenesis in JZ-10 and YLL-10 to further verify the observation results obtained from paraffin sections and scanning electron microscopy.

Figure 7 displays cross-sections of florets from top to bottom successively, reflecting the trend of differentiation of the JZ co primordia. Discontinuous upper margins of co were first found in JZ-A, with c1a and c1b and c2a and c2b on both sides of the point where the second whorl of stamen filaments (fi2) bound to the inner wall of the hypanthium and the third pair of c3a and c3b in the slightly downward section in JZ-B, demonstrating that fi2 blocks the differentiation path of co primordia to some extent at the early stage of co primordia differentiation. In the lower section of JZ-C, the co primordia had three complete segments on the outer side of fi2, with notable notches between them just on the outer side of the first whorl of stamen filaments (fi1). As differentiation progressed and the segment between the top of the hypanthium and fi2 elongated and grew, the co primordia soon broke through the blockage from fi2 and fused together, but the blockage from fi1 to the differentiation of co was still present. This result is in line with the small notches observed by scanning electron microscopy. In the lowermost section of JZ-D, the blockage from fi1 was also successfully broken through, eventually forming a closed-loop co structure. The above results suggest that the JZ co primordia experienced three-stage differentiation, i.e., blockage from the second whorl of stamen, blockage from the first whorl of stamen block, and healing of co primordia.

It was discovered that c4 was found between fi1 and anther locules (po) in JZ-A, c5 was observed in JZ-B, and c3b was delayed in occurrence, indicating that the differentiation of corona primordia is also affected by factors other than the influence of staminal filaments. In JZ-B, there were vacancies between c4 and c3a and between c5 and c2b, suggesting that the anther locules po also interfere with the differentiation of corona primordia. This interference differs from the blockage from the point where staminal filaments bound to the inner wall of the hypanthium and may be ascribed to physical compression. The above phenomenon also signifies that corona primordia differentiation occurs at multiple sections and corona primordia are continuously distributed despite the effect of stamen morphology.

The effect of petal-like stamens on the differentiation of corona primordia was also found in YLL-10. As shown in Figure 8, A−F show the trend of differentiation of YLL corona primordia. In YLL-A and YLL-B, the typical Co primordia were first observed in ST2, with fragmented Co primordia at two points. In YLL-C, suspected Co primordia were also found in ST1, but in YLL-D, they turned out to be the extension of Co adjacent to ST2. At this point, the base of ST1 was inward, probably giving way to the differentiation of Co primordia, thereby resulting in the extension of Co adjacent to ST2. It was further observed in YLL-E that typical and fragmented Co primordia healed at the base. In YLL-F, the Co primordia had an overall healing trend. The above observations suggest that spatial competition at the base of the petal-like stamen ST1 is the leading cause of abnormal Co differentiation.

More details of high resolution X-ray topography for JZ-10 and YLL-10 can be found in Appendix A, respectively.

## 3. Discussion

The analysis results showed that corona primordia appeared at the late stage of floral organ differentiation, in line with previous findings [8,15,17,18]. The specificity of the corona differentiation stage is one piece of the evidence showing that corona is a structure independent of floral organs.

The development process of the corona was accompanied by the elongation and growth of the hypanthium, and the morphogenesis of the corona appeared to be inextricably linked to the hypanthium. After the hypanthium had differentiated and started to elongate, the corona primordia of JZ began to differentiate on the inner wall of the juncture of the base of tepals and the upper margin of the hypanthium, and the differentiation was continuous. However, it was uncovered through scanning electron microscopy and high-resolution X-ray tomography that this continuity was interrupted by filaments that did not elongate and grow. The staminal filaments seemingly blocked the differentiation path of the corona, but a complete ring was still formed by the corona primordia that differentiated soon at the blockage and the rest of the corona. As the hypanthium and staminal filaments elongated and grew, the distance between the top of the hypanthium and the base of the second whorl of staminal filaments gradually lengthened, the blockage from the two whorls of stamens to corona differentiation was successively eliminated, and the path of corona differentiation below small notches was healed. Thus, the differentiation of the JZ corona can be divided into three stages: blockage from the second whorl of staminal filament (I), blockage from the first whorl of stamen filament (II), and healing of corona primordia (III) (Figure 9). The corona primordia seemed to discontinuously differentiate due to blockage from staminal filaments and anther loculus extrusion, but the differentiation of the corona of JZ seemed to be controlled by certain “genes” for continuous differentiation. Waters et al. [8] also found high points in the corona between stamens in paraffin cross-sections of *Narcissus bulbocodium* and thus believed that the corona differentiates from six symmetrical points of primordia between stamens and tepals, clearly inconsistent with the findings on Chinese *Narcissus*, i.e., the corona primordia of Chinese *Narcissus* have continuous differentiation sites.

The continuity of the differentiation of corona primordia is determined by the development processes of both stamens and the hypanthium. Similar to that of JZ, the differentiation site of the corona primordia of YLL was also located on the inner wall of the juncture of the base of tepals and the upper margin of the hypanthium. The difference lay in that the differentiation of the corona primordia of YLL was discontinuous. The half-surrounded typical corona primordia first differentiated at the base of the second whorl of petal-like stamens. Then, bump-like fragmented corona primordia differentiated at the interval between the two whorls of petal-like stamens. Later, fragmented corona primordia differentiated at the base of the first whorl of petal-like stamens, and typical and fragmented corona primordia were eventually combined at the base in a circle, forming a fragmented corona with a discontinuous upper margin. In terms of the discontinuous differentiation of the fragmented corona, the anatomical evidence suggests that the base of the second whorl of petal-like stamens was located in the middle of the hypanthium, indicating that the upper part of the hypanthium elongated and grew, and that corona primordia broke through the blockage from the second whorl of petal-like stamens, signifying that corona primordia can complete normal differentiation and development. For the first whorl of petal-like stamens inserted on the top of the hypanthium, the anatomical evidence revealed that the junction extent between the base of the petaloid stamens and the hypanthium was larger than filament of JZ, thereby blocking or hindering the differentiation path of corona primordia. As a result, the differentiation of the corona primordia at the base of the first whorl of petal-like stamens was delayed, and the corona primordia on both sides were bumpy and misshapen. Although in the longitudinal section of YLL at the full-bloom stage the hypanthium at the upper part of the second petal-shaped stamen also did not significantly elongate (Figure 1, YLL-B), the paraffin section observation showed that the second whorl of the petalized stamen reserved a larger space for the differentiation of corona compared to the first whorl (Figure 4, YLL-8). This is supported by the morphology of the YLL corona at the full-bloom stage, where there were outermost three large and three small, fragmented coronas (Figure 10). Therefore, the development pattern of the YLL corona may be that after completing JZ-co-I of JZ, spatial occupancy caused by petalization of stamens led to the failure of the corona primordia to break through the first whorl of petal-like stamens and eventually the formation of fragmented corona.

Corona differentiation clearly received a positive or negative influence from the stamens in terms of spatial position. Staminal petalization led to fragmented corona. Additionally, at the full-bloom stage, petal-like stamens of YLL had a petal-like upper part and a corona-like lower part, which was almost the same color as corona (Figure 1, YLL-A). The formation mechanism of stamen petaloid may be another key to unlocking the origin of corona completely.

This study suggests that the spatio-temporal specificity of corona differentiation is a strong proof that the corona exists independently of typical floral organs such as tepals, stamens, and gynoecium. Stamens exert an evident effect in corona differentiation, and the two may have a closer relationship. In future studies, the regulatory mechanism of the morphogenesis of the Chinese *Narcissus* corona can be explored at the molecular level.

## 4. Materials and Methods

### 4.1. Plant Materials and Storage Conditions

The flower buds and the largest floret in the inflorescences of the main bulbs of the three-year-old commercial bulbs of JZ and YLL were used as materials. The bulbs of both cultivars were produced in Zhangzhou, Fujian Province, China and sorted according to the Grade 3 standard [19] in early June in 2021 to remove diseased and damaged bulbs. In late July, the materials were transported to Chaoyang District, Beijing, China, and stored in a shade with natural ventilation. COS-03-X was employed to record changes in temperature and humidity. During storage, the average daily humidity was kept above 50%, and the maximum and minimum daily temperatures were maintained below 30 °C and at 20–25 °C, respectively. Sampling was carried out from 31 July to late October, during which observation was made every 6 d.

### 4.2. Observation Using a Stereomicroscope

Inflorescences and florets were dissected with hands using dissecting needles or forceps under a stereomicroscope (Stemi 305) to observe corona development, and images were acquired (Axiocam ERc 5s).

### 4.3. Paraffin Section Preparation

Based on stereomicroscopic observations, florets in the early, middle, and late stages of corona differentiation were taken and fixed in an FAA fixative solution (G1108, 50% alcohol). The paraffin sections were prepared through the following 8 steps: (1) dehydration and soak in paraffin: tissues and the corresponding labels were placed in a dehydration box. Then, the dehydration box was placed in a dehydration machine (Donatello) for dehydration in 75% alcohol for 4 h, 85% alcohol for 2 h, 90% alcohol for 2 h, 95% alcohol for 1 h, and anhydrous ethanol (100092683) for 30 min 2 times. Next, it was subjected to treatment with alcohol benzene for 5–10 min and then xylene (10023418) for 5–10 min 2 times. After that, paraffin was melted at 65 °C, and the tissues were soaked in the paraffin for 1 h, with the melted paraffin replaced 2 times; (2) embedding: an embedding machine (JB-P5) was used for embedding. In brief, the melted paraffin was firstly put into an embedding frame, and tissues were removed from the dehydration box, placed in the embedding frame and labelled according to the requirements of the embedding surface prior to the solidification of the melted paraffin. Thereafter, they were subjected to cooling on a freezing table (JB-L5) at −20 °C. Following solidification, the paraffin block was taken out from the embedding frame and trimmed; (3) sectioning: the paraffin block trimmed was placed on a freezing table for cooling at −20 °C. Then, the cooled paraffin block was put into a microtome (RM2016) to prepare sections with a thickness of 4 μm. Thereafter, tissues were flattened at 40 °C on warm water in a spreader (KD-P) and picked up with a slide. Afterwards, sections were baked in an oven (GFL-230) at 60 °C and then stored at room temperature; (4) deparaffinization of paraffin sections: paraffin sections were subjected to deparaffinization. Specifically, sections were sequentially placed in environmentally friendly deparaffinizing transparent liquid (G1128) 2 times for 20 min, anhydrous ethanol 2 times for 5 min, and 75% alcohol 1 time for 5 min and finally rinsed with tap water; (5) Safranin staining: sections were stained in a plant saffron staining solution (G1031) for 2 h and washed with tap water to remove excess dye; (6) de-staining: sections were placed in 50%, 70%, and 80% gradient alcohol for 3–8 s each; (7) fast green staining: sections were stained in a plant solid green staining solution for 6–20 s and dehydrated in anhydrous ethanol 3 times; (8) permeabilizing and sealing: sections were permeabilized in xylene for 5 min and sealed with neutral gum (10004160). Finally, sections were observed using a microscope (Primo Star), and images were acquired (Axiocam 208 color).

### 4.4. Scanning Electron Microscopy

Intact florets in the middle and late stages of corona differentiation were stripped and put into a 1 mL centrifuge tube containing an electron microscope fixative (G1102), followed by standing at room temperature for 2 h and then storage at 4 °C. For some samples, the outer whorl of tepals and stamens or petal-like stamens and longitudinally cut florets should be stripped in advance. Next, the samples were prepared as follows: (1) The centrifuge tube was centrifuged at 3000 rpm to remove the fixative glutaraldehyde, followed by dehydration with ethanol at different concentrations (30%, 50%, 70%, 80%, 90%, 95%, and 100%) 2 times (15 min/time) at each concentration. (2) The samples were dried through the critical-point drying method (Autosamdri-815). In brief, liquid carbon dioxide was filled in to displace ethanol for 20–60 min. Then, the temperature and pressure were increased until reaching the critical point of carbon dioxide and maintained for 4 min. Afterwards, the carbon dioxide was slowly released for approximately 30 min while maintaining the temperature and pressure, and the samples were taken out. (3) The samples were subjected to conductive treatment by the vacuum spraying method (MC1000). Specifically, the samples were placed on a sample table 10–15 cm from an evaporation source, rotated and uniformly sprayed gold (10 KV, 1000 s); (4) observation was carried out through a scanning electron microscope SU 8020. The parameters of SU 8020 were as follows: energy spectrum: HORIBA EX250; electron gun: a cold field emission source; accelerating voltage: 0.5–30 kV (standard mode); landing voltage: 0.1–2.0 kV (deceleration mode); low-power mode: 20–2000× (image magnification); high-power mode: 100–800,000× (image magnification); secondary electron image resolution: 1.0 nm (accelerating voltage: 15 kV, WD = 4 mm) and 1.3 nm (landing voltage: 1 kV, WD = 1.5 mm).

### 4.5. High-Resolution X-ray Tomography

The florets in the middle and late stages of corona differentiation to be observed were peeled from inflorescences and placed vertically in a 0.2 mL centrifuge tube with the anthophore downward. Next, the tube was capped to prevent excessive water loss, and the excess of the cap was trimmed off, so that the cap was the same as the mouth of the tube in diameter. Afterwards, the base of the tube was fixed vertically to the sample table of a high-resolution X-ray tomography scanner (SKYSCAN 2214) with special paraffin. Thereafter, scanning was conducted following adjustment of the distance and parameters. The scanning parameters were set as follows: a resolution of 2.101213 μm, a voltage of 46 kV, a current of 90 μA, and a time of exposure of 180 ms.

## Figures and Tables

**Figure 1 plants-12-01458-f001:**
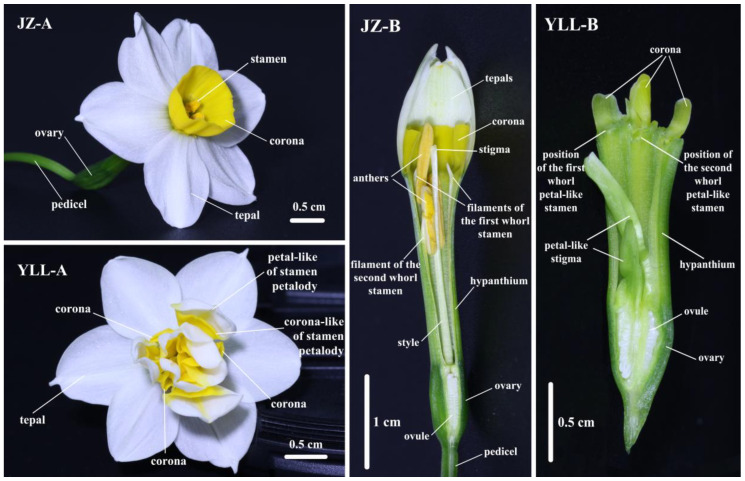
Flower types of JZ and YLL. JZ and YLL denote “Jinzhanyintai” and “Yulinglong”, respectively. A represents flower morphology at the full-bloom stage, and B represents the longitudinal section of flower.

**Figure 2 plants-12-01458-f002:**
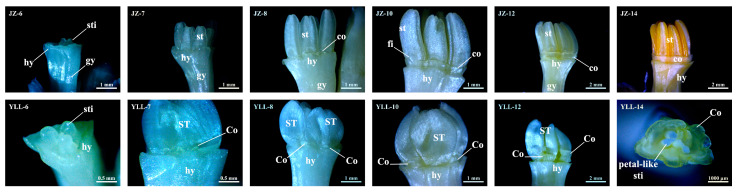
Images showing the corona differentiation of JZ and YLL by dissection with hands. JZ and YLL denote “Jinzhanyintai” and “Yulinglong”, respectively, and numeric characters represent sampling periods. For example, JZ-1 denotes the first observation for JZ in July 31. JZ-7 to JZ-14 and YLL-7 to YLL-12 represent tepals stripped for JZ and YLL in respective periods, JZ-6 represents the observation of tepals and stamens stripped for JZ in period 6, and YLL-6 and YLL-14 represent tepals and petal-like stamens stripped for YLL in periods 6 and 14, respectively. hy—hypanthium; gy—gynoecium; sti—stigma; st—stamen; co—corona; fi—filament; ST—petal-like stamen; Co—fragmented corona.

**Figure 3 plants-12-01458-f003:**
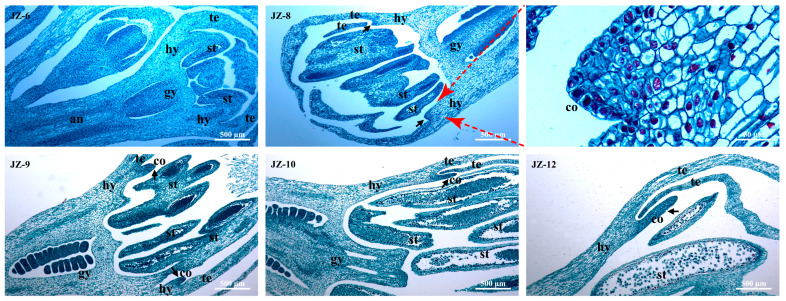
Longitudinal paraffin sections of the JZ corona during morphogenesis. an—anthophore; hy—hypanthium; gy—gynoecium; te—tepal; st—stamen; co—corona.

**Figure 4 plants-12-01458-f004:**
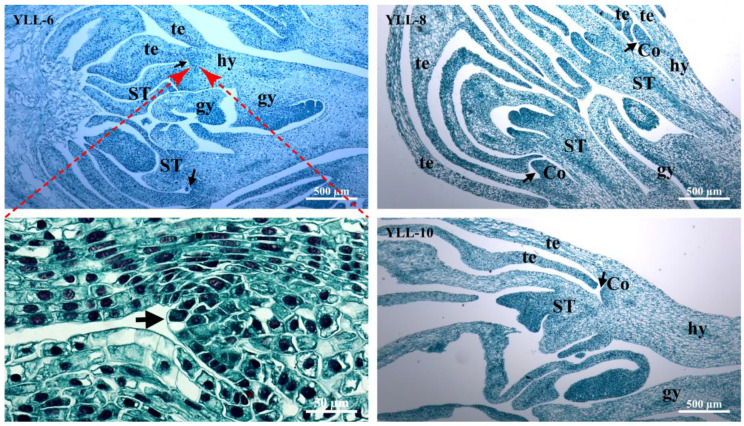
Longitudinal paraffin section of the YLL fragmented corona during morphogenesis. hy—hypanthium; gy—gynoecium; te—tepal; ST—petal-like stamen; Co—fragmented corona.

**Figure 6 plants-12-01458-f006:**
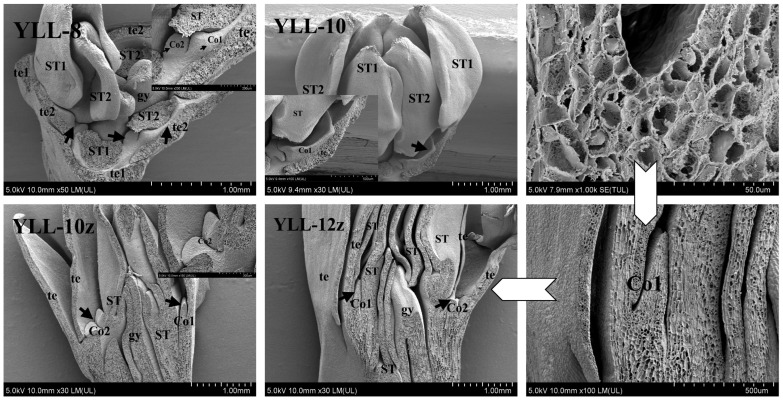
Scanning electron microscopic images of YLL fragmented corona during morphogenesis. hy—hypanthium; gy—gynoecium; te—tepal; ST—petal-like stamen; Co—fragmented corona. YLL-8 and YLL-10 represent all tepals stripped for YLL in periods 8 and 10, respectively; YLL-10z and YLL-12z: longitudinal sections for YLL in periods 10 and 12, respectively.

**Figure 7 plants-12-01458-f007:**
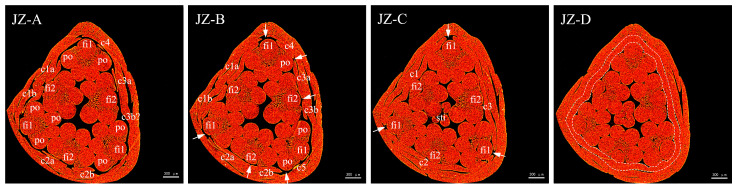
X-ray cross-sectional tomographic images of morphology of the JZ corona. JZ-A to JZ-D are cross-sections in order from top to bottom with the spacings of 25 × 2.1 μm, 64 × 2.1 μm, and 35 × 2.1 μm, respectively, where c1a, c1b, c1, c4, and the like denote corona primordia, fi1 and fi2 stand for filaments of the first and second whorls of stamens, respectively, po represents anther locules, and sti denotes stigma. White dashed lines in JZ-D denote corona primordia that formed a complete structure.

**Figure 8 plants-12-01458-f008:**
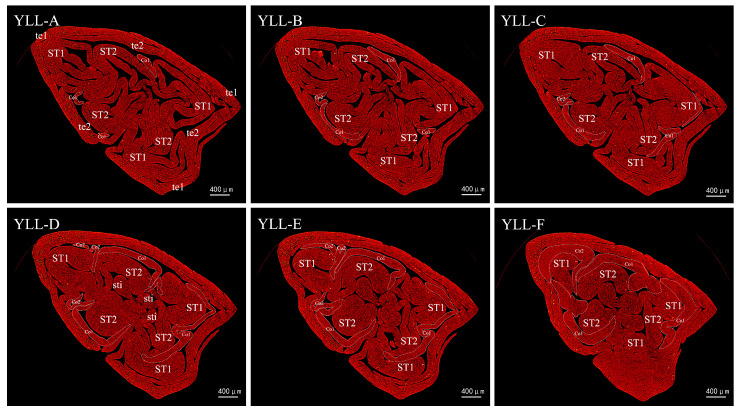
X-ray cross-sectional tomographic images of morphology of the YLL corona. YLL-A to YLL-F are cross-sections from top to bottom in order with the spacings of 46 × 2.1 μm, 31 × 2.1 μm, 126 × 2.1 μm, 95 × 2.1 μm, and 156 × 2.1 μm, respectively, where the white dashed circles denote corona primordia in all sections, and ST1 and ST2 denote the first and second whorl of petal-like stamens, respectively.

**Figure 9 plants-12-01458-f009:**
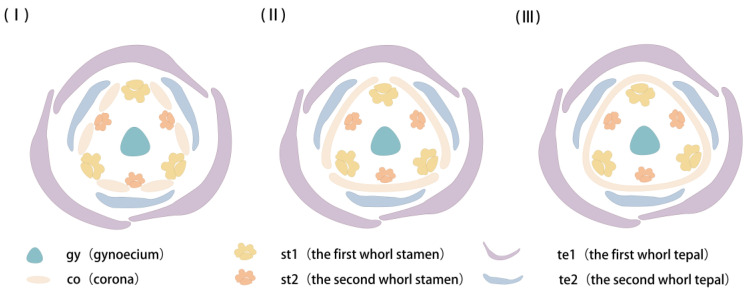
Development model of the JZ corona. te1 and te2 denote the first and second whorls of tepals, respectively, st1 and st2 denote the first and second whorls of stamens, respectively, gy denotes gynoecium, the yellow patterning denotes the corona in morphogenesis, and (**I**–**III**) denote the three stages of corona development.

**Figure 10 plants-12-01458-f010:**
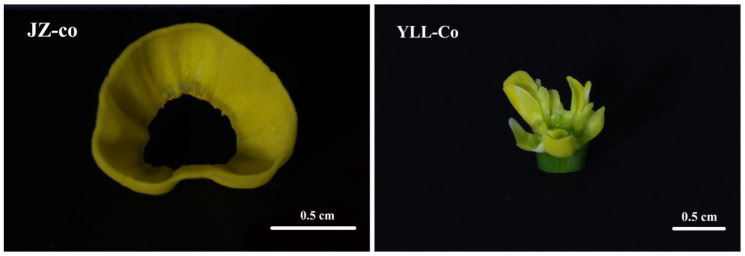
Anatomical morphologies of the corona of JZ and YLL at the full-bloom stage. JZ-co—JZ corona; YLL-Co—YLL corona.

## Data Availability

The datasets that support the findings of this study are available from the corresponding author on reasonable request.

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
