# Peer review of "Emergence of Corona Is Independent of the Four Whorls of Floral Organs in *Narcissus tazetta"

_plants, 2023, doi:10.3390/plants12071458_

Round 1
Reviewer 1 Report
This manuscript describes the histological structure of the developing flowers of Narcissus Tazetta. To explore the formation of corona structure they compare two cultivars with fused or nonfused corona.
They observe that the corona is the last floral structure to be initiated and arises from the specific holder for external floral organizing the hypanthium as observed previously.
From this observation, they concluded that the corona is not derived from the classic floral organs (needer stamens nor tepals). Their growth is related to the space between tepals and stamens. That the expansion of the stamens during pollen sacs growth relates to corona growth and fusion.
This is an interesting work that gives the bases for further studies including transcriptional data.
As it stands it's a small increment but an important one with the previous work. Soch's work was always considered to be sufficient for publication yet today most journals and readers expect more genetic data.
I found the figures clear and the data well presented and the writing clear.
All the conclusions are suitable and justified.
I can only regret that the transcriptional data is not included in this work. A full understanding of the origin of the corona requires an exact description of the transcription factors involved in its differentiation. This data could greatly change the vision given by the cytological arguments presented in this paper.
Reviewer 2 Report
The manuscript "plants-2294245" is a well-illustrated and well-structured descriptive study, focusing on the anatomical identity of the corona, which is typical of Narcissus flowers. Taking advantage of the differences among two varieties and applying an appropriate assortment of methodology, the authors reach to a convincing conclusion: "the corona is a structure independent of and different from the typical four whorls of floral organs".
The overall research design is elegant and only a few points have to be corrected:
1. Although the text is well-written and reads easily, especially the abstract is very badly presented, only describing the findings and final conclusion. So please, rewrite the abstract with decent English syntax and a rationale that explains the whole concept at a "miniature": background, methodology, key results and conclusion.
2. In the title please write: "Emergence of Corona is Independent....."
3. Figures 1 and 2 require labelling. Could you supply a higher magnification of a typical flower, labelling the parts of it?
4. In general, the "Note" parts have to be incorporated in the figure legends.
5. Because Materials and Methods come after the Results, it is difficult to understand the dates and periods stated at the beginning of Results. So, if possible, provide Materials andd Methods before the Results. Otherwise, give in a short paragraph the "timetable" of sampling and proccessing for this study.
